# CO-tolerant RuNi/TiO₂ catalyst for the storage and purification of crude hydrogen

Zhaohua Wang[1,5], Chunyang Dong[1,2,5], Xuan Tang[3,5], Xuetao Qin[1], Xingwu Liu ®[1,4], Mi Peng[1], Yao Xu[1], Chuqiao Song[1], Jie Zhang[1], Xuan Liang[1], Sheng Dai ®[3] ✉ & Ding Ma ®[1] ✉

Hydrogen storage by means of catalytic hydrogenation of suitable organic substrates helps to elevate the volumetric density of hydrogen energy. In this regard, utilizing cheaper industrial crude hydrogen to fulfill the goal of hydrogen storage would show economic attraction. However, because CO impurities in crude hydrogen can easily deactivate metal active sites even in trace amounts such a process has not yet been realized. Here, we develop a robust RuNi/TiO₂ catalyst that enables the efficient hydrogenation of toluene to methyl-cyclohexane under simulated crude hydrogen feeds with 1000–5000 ppm CO impurity at around 180 °C under atmospheric pressure. We show that the co-localization of Ru and Ni species during reduction facilitated the formation of tightly coupled metallic Ru-Ni clusters. During the catalytic hydrogenation process, due to the distinct bonding properties, Ru and Ni served as the active sites for CO methanation and toluene hydrogenation respectively. Our work provides fresh insight into the effective utilization and purification of crude hydrogen for the future hydrogen economy.

Hydrogen, one of the most promising energy vectors, its practical storage and transportation have always been the bottleneck problem against the further development of hydrogen economy[1–3]. Compared with physical-based approaches such as compressing or liquefying molecular hydrogen at extreme conditions, chemical hydrogen storage utilizing catalytic hydrogenation of organic substrates to the so-called liquid organic hydrogen carrier (LOHC) has been regarded as a more safe and alternative way[4–6]. Among the reported LOHC candidates, toluene and corresponding methyl-cyclohexane (MCH) constitute a promising medium for hydrogen storage due to their low-toxicity, superior reversibility and stability, high gravimetric hydrogen capacity (6.2 wt%), and facile hydrogen charge/discharge property[7]. For the traditional catalytic hydrogenation process, the usage of pure hydrogen (over 99.99%) is indispensable; otherwise, only a trace

amount of CO impurity in hydrogen can easily deactivate the metal active sites[8,9]. However, as for the industrially crude hydrogen resources, namely those obtained from steam reforming and water-gas shift reactions with a small amount of CO as the main contamination, have only half the price of the pure hydrogen[10,11], making the development of CO-tolerant hydrogenation catalysts attractive in both economic values and scientific significance.

The strong bonding formation between CO and the commonly used metal hydrogenation catalysts are reinforced by the transfer of the d-electrons from metal to the 2π* anti-bonding orbital of CO molecule[9]. Previous attempts showed that by decreasing the size of metal to atomic scale, the resulting cationic metal single atoms with less d-electrons tend to bind CO in a less intensive manner[12–15]. As for the hydrogenation of aromatic substrate which generally requires

[1]Beijing National Laboratory for Molecular Sciences, College of Chemistry and Molecular Engineering, Peking University, Beijing 100871, P. R. China. [2]UCCS–Unité de Catalyse et Chimie du Solide, Université de Lille, CNRS, Centrale Lille, ENSCL, Université d'Artois, UMR, 8181 Lille, France. [3]Key Laboratory for Advanced Materials and Joint International Research Laboratory of Precision Chemistry and Molecular Engineering, Feringa Nobel Prize Scientist Joint Research Centre, Frontiers Science Center for Materiobiology and Dynamic Chemistry, Institute of Fine Chemicals, School of Chemistry and Molecular Engineering, East China University of Science & Technology, Shanghai 200237, China. [4]National Energy Center for Coal to Liquids, Synfuels CHINA Co., Ltd, Beijing 101400, China. [5]These authors contributed equally: Zhaohua Wang, Chunyang Dong, Xuan Tang. ✉e-mail: shengdai@ecust.edu.cn; dma@pku.edu.cn

metal ensemble sites[16–20], Su et al. recently discovered that with the assistance of neighboring oxygen vacancies, Pt single atom catalyst manifests even higher toluene hydrogenation performance than catalysts with Pt nanoparticles (NPs)[21]. Despite the theoretical potential of single atom catalysts in driving the CO-tolerant toluene hydrogenation process, their attenuated interaction toward CO would inevitably compromise the purification capabilities toward the crude hydrogen feed. Alternatively, the parallel CO methanation along with toluene hydrogenation offers a more practical solution to alleviate the CO-poisoning effect of the metal to achieve simultaneous crude hydrogen storage and purification. To this end, the fabrication of catalysts with bifunctional active sites for dual hydrogenation paths would be highly necessary. Yet to date, no such catalytic system has ever been reported.

Herein, we develop a robust RuNi/TiO$_2$ catalyst with both CO methanation and toluene hydrogenation performance, bringing the concept of crude hydrogen storage and purification into reality. Using crude hydrogen (0.5%CO/H$_2$, v/v) and toluene vapor (WHSV = 1.4 h$^{-1}$) as the inlet feed, the optimized catalyst maintained over 60% MCH yield in 24 h reaction and reduced the concentration of CO in the outlet hydrogen stream down to 0.01%. Combined with atomic-resolved microscopic and spectroscopic studies, we show that the formation of tightly coupled Ru–Ni clusters is the key to simultaneously realizing the two separated hydrogenation reaction paths. Based on their distinct binding properties, Ru and Ni served as the active sites for CO methanation and toluene hydrogenation, respectively.

## Results

### Catalytic storage and purification of crude hydrogen

The TiO$_2$-supported Ru, Ni, and bimetallic RuNi catalysts were prepared by the incipient wetness impregnation (IWI) method, denoted as 2Ru/TiO$_2$, 5Ni/TiO$_2$, and 2Ru$x$Ni/TiO$_2$ (0.5 ≤ $x$ ≤ 8) based on the weight percent of each metallic component on the catalyst (see "Methods"). Other metallic catalysts we used are all commercial ones. The evaluation of different catalysts in toluene hydrogenation with simulated crude hydrogen as carrier gas was performed in a fixed bed flow reactor at atmospheric pressure (Supplementary Fig. 1). According to literature reports, transition metal NPs such as Ru, Pd, and Pt are highly reactive for the hydrogenation of aromatics[22–25]. However, along with the addition of 1000 ppm CO in the feed hydrogen, the rate of toluene conversion of these catalysts were sharply suppressed at 170 °C (Fig. 1a). Even in a wide temperature range from 140 to 200 °C, all the commercial catalysts are almost inactive toward CO-tolerant toluene hydrogenation (MCH yields < 10%, Supplementary Fig. 2). Remarkably, after combining Ru and Ni together, the resulted 2Ru5Ni/TiO$_2$ manifested significantly elevated toluene conversion rate of 12.5 mol$_{toluene}$/mol$_{metal}$/h (moles of Ru and Ni were both included in the calculation) and MCH yields (20–60%), demonstrating the feasibility of crude hydrogen storage (Fig. 1a and Supplementary Fig. 2). Specifically, under a temperature-programmed toluene hydrogenation profile from 100 to 220 °C, the CO-tolerant hydrogenation performance of 2Ru5Ni/TiO$_2$ initiated from 120 °C, after reaching the highest MCH yield of 58.1% at 170 °C, the activity dropped gradually with the elevation of temperature to 220 °C (Fig. 1b). Regardless of the temperature variations, MCH was always the dominant hydrogenation product (selectivity > 99%) from toluene (Supplementary Fig. 3). Repeating the catalytic evaluation of 2Ru5Ni/TiO$_2$ under the same temperature ramp manifested a similar profile (Supplementary Fig. 4), suggesting the activity decrease at higher temperature did not result from its deactivation (Supplementary Note 1). Moreover, by varying the loadings of the Ru and Ni of the catalyst, the current 2Ru5Ni/TiO$_2$ turned out to be the catalyst with the optimal performance, and higher loadings of Ru and Ni have not contributed to higher MCH yields (Fig. 1b and Supplementary Fig. 5). To understand the reason why bimetallic 2Ru5Ni/TiO$_2$ overwhelms its monometallic counterparts in the catalytic crude

hydrogen storage, CO conversion in the above crude toluene hydrogenation reaction of representative samples of 2Ru/TiO$_2$, 5Ni/TiO$_2$, 2Ru5Ni/TiO$_2$, and their MCH yields in pure hydrogen were evaluated separately (Fig. 1c, d). Apparently, both 2Ru/TiO$_2$ and 2Ru5Ni/TiO$_2$ possessed superior low-temperature CO methanation activities, in which over 95% CO could be transformed into methane at 170 °C and above. In contrast, the 5Ni/TiO$_2$ only managed 20% CO conversion at 170 °C (CO conversion profiles for other 2Ru$x$Ni/TiO$_2$ catalysts are shown in Supplementary Fig. 6). As for the toluene hydrogenation evaluation with pure hydrogen, surprisingly, completely different temperature-activity profiles were observed for 2Ru/TiO$_2$ and 2Ru5Ni/TiO$_2$. When the reaction temperature increased to 160 °C and above, only less than 20% MCH yields was left for 2Ru/TiO$_2$, and this marks a huge difference from the high MCH yields (>60%) of 2Ru5Ni/TiO$_2$ from 120 to 190 °C. Such significant activity loss of 2Ru/TiO$_2$ with the temperature increasing can be ascribed to the gradual diminish of surface coverage of toluene or hydrogen due to desorption[26,27], which can also explain the volcanic curve in temperature-dependent MCH yields for 2Ru5Ni/TiO$_2$ (Fig. 1b, d and Supplementary Fig. 5). Apparently, the incompatibility for the parallel reactions in temperatures causes 2Ru/TiO$_2$ far less reactive than 2Ru5Ni/TiO$_2$ during the above crude hydrogen storage process.

To further examine the capability of 2Ru5Ni/TiO$_2$ against CO deactivation, under a lower WHSV of toluene = 1.4 h$^{-1}$, the toluene hydrogenation reactions were performed using crude hydrogen carrier gas with different CO concentration (Fig. 1e). It was found 2Ru5Ni/TiO$_2$ remained active with considerable MCH yields when CO was below 0.5%. Specifically, the MCH yields were 91% in 0.1%CO/H$_2$ and 73% in 0.5% CO/H$_2$, with 100% CO conversion. Further increasing the concentration of CO led to the incomplete conversion thus causing a moderate drop of MCH yield (MCH yield of 29% in 1%CO/H$_2$). Therefore, 0.5% CO in H$_2$ is a bearable concentration with a great capability of 2Ru5Ni/TiO$_2$ to convert toluene and CO simultaneously in this condition.

The 2Ru5Ni/TiO$_2$ catalyst also represented catalytic robustness in the long-term reaction test. Significantly, after over 24 h test on stream, the MCH yields maintained over 60% while reducing the concentration of CO in the outlet hydrogen stream down to 0.01% (Fig. 1f). Notably, since the Ni contents kept unchanged after the test, the potential formation of poisonous Ni(CO)$_4$ thus could be excluded (Supplementary Table 1). However, the slight MCH yield drop from starting 75% to final 60% may be ascribed to the metal particle agglomeration during the reaction, as the Ru and Ni growth were observed through following scanning transmission electron microscopy (STEM) characterization of spent 2Ru5Ni/TiO$_2$ catalyst (Supplementary Fig. 10). For large scale application, trickle bed reactor is more suitable and the result using toluene liquid by pump is shown in Supplementary Fig. 7. The conversion could maintain 22–36% during the test. In addition, to prove the developed catalyst is not limited to toluene, we extended the substrates using both gas-solid fixed bed reactor and batch reactor (Supplementary Fig. 8 and Supplementary Table 2). As the results show, 2Ru5Ni/TiO$_2$ could efficiently realize CO-tolerant hydrogenation of these aromatics, suggesting the wide feasibility of the catalyst. Moreover, under more practical circumstance that the crude hydrogen feeds contain both CO and CO$_2$, the relatively low reaction temperature of 2Ru5Ni/TiO$_2$ in the present case guaranteed that no hydrogen would be wasted for CO$_2$ methanation (Supplementary Fig. 9)[28–30].

### Structural characterization of catalysts

To explore the exceptional performance of 2Ru5Ni/TiO$_2$ in the crude hydrogen storage process, we investigated the structural evolution of Ru and Ni species during different states of the synthesis using aberration-corrected scanning transmission electron microscopy (AC-STEM) and energy-dispersive X-ray spectroscopy (EDS). All the

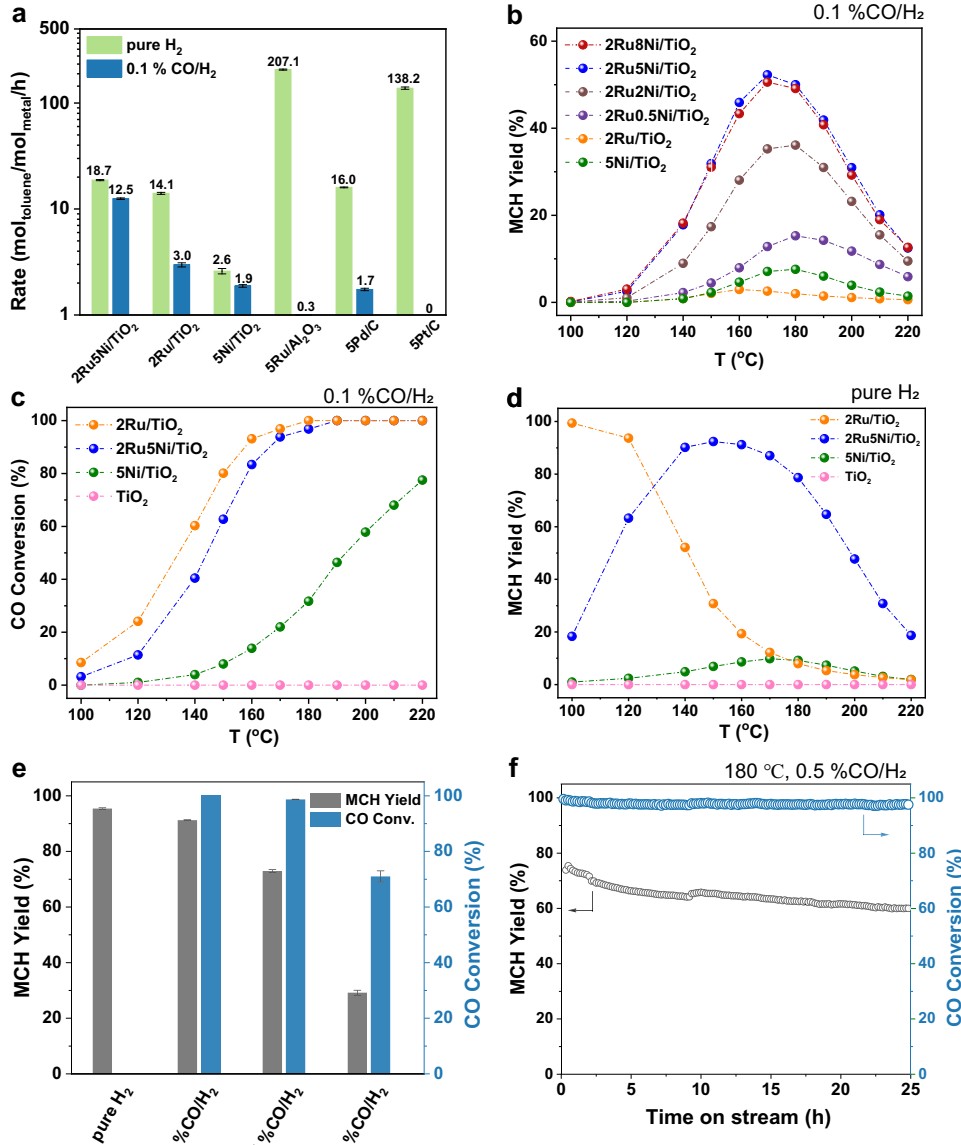

**Fig. 1 | Toluene hydrogenation performance of catalysts under different conditions. a** The metal-normalized toluene conversion rates at 170 °C over different catalysts using either pure hydrogen or 0.1%CO/$H_2$ (CO:Ar:$H_2$ = 0.1:0.1:99.8, v/v, GHSV (gas hourly space velocity) = 36,000 mL/$g_{cat}$/h) as carrier gases. The conversion rates of toluene were measured in kinetic region (i.e., toluene conversion <15%). **b** MCH yields of different catalysts in 0.1% CO/$H_2$ carrier gas. **c** CO conversion of different catalysts in 0.1% CO/$H_2$ carrier gas with toluene. **d** MCH yields of different catalysts in pure hydrogen carrier gas. The measurements for

(**b**)–(**d**) were performed in the conditions: GHSV of the carrier gas = 36,000 mL/$g_{cat}$/h and WHSV (weight hourly space velocity) of toluene = 2.1 h$^{-1}$. **e** MCH yields and CO conversion of 2Ru5Ni/$TiO_2$ at 180 °C in hydrogen carrier gas with various CO concentrations (0–1%). **f** Stability test of 2Ru5Ni/$TiO_2$ at 180 °C using 0.5%CO/$H_2$ as carrier gas. Reaction conditions for (**e**) and (**f**): GHSV of the carrier gas = 12,000 mL/$g_{cat}$/h and WHSV of toluene = 1.4 h$^{-1}$. Error bars for the activity in (**a**) and (**e**) represent the standard deviation from at least three independent measurements.

reduced samples were prepared in a glove box under an Ar atmosphere and then transferred to a transmission electron microscope using a vacuum transfer holder to eliminate air exposure. To reduce the possible beam damage on the catalysts, an integrated differential phase contrast (iDPC)-assisted high angle annular dark field (HAADF)-STEM imaging approach was utilized (more details can be found in "Methods"). First, prior to the calcination, the co-impregnation of Ru and Ni on $TiO_2$ led to the formation of the atomically distributed Ru and Ni species in an evenly mixed state (Supplementary Fig. 11). After calcination, $RuO_2$ species formed which were strongly trapped on the $TiO_2$ surface due to the epitaxial lattice matching between $RuO_2$ (110) (3.23 Å) and rutile $TiO_2$ (110) (3.28 Å) in both 2Ru/$TiO_2$ and 2Ru5Ni/$TiO_2$ (Fig. 2a, b and Supplementary Fig. 12)[31,32]. The obvious co-localization between Ru and Ni species was also observed in calcined 2Ru5Ni/$TiO_2$,

suggesting a strong interaction between the NiO and $RuO_2$ species (Fig. 2a, b and Supplementary Fig. 13). From the HAADF-STEM images and the corresponding EDS elemental maps of reduced 2Ru5Ni/$TiO_2$ (Fig. 2c, d and Supplementary Fig. 14), both Ru and Ni species kept their high dispersion states. Though atomic-scale HAADF-STEM image of 2Ru5Ni/$TiO_2$ cannot be acquired due to the strong magnetism after reduction, it is reasonable that the epitaxial lattice matching still exists because the strongly trapped Ru species on the $TiO_2$ can also be observed in 2Ru5Ni/$TiO_2$ (Fig. 2c, d). Specifically, most Ni species which had a smaller size distribution of 2.3 ± 1.5 nm were adjacent to Ru atoms (Supplementary Fig. 15). By contrast, in 5Ni/$TiO_2$, Ni species were highly dispersed on $TiO_2$ after calcination (Supplementary Fig. 16) but agglomerated violently after reduction with a 65.8 ± 29.3 nm size distribution (Fig. 2e, f and Supplementary Fig. 17).

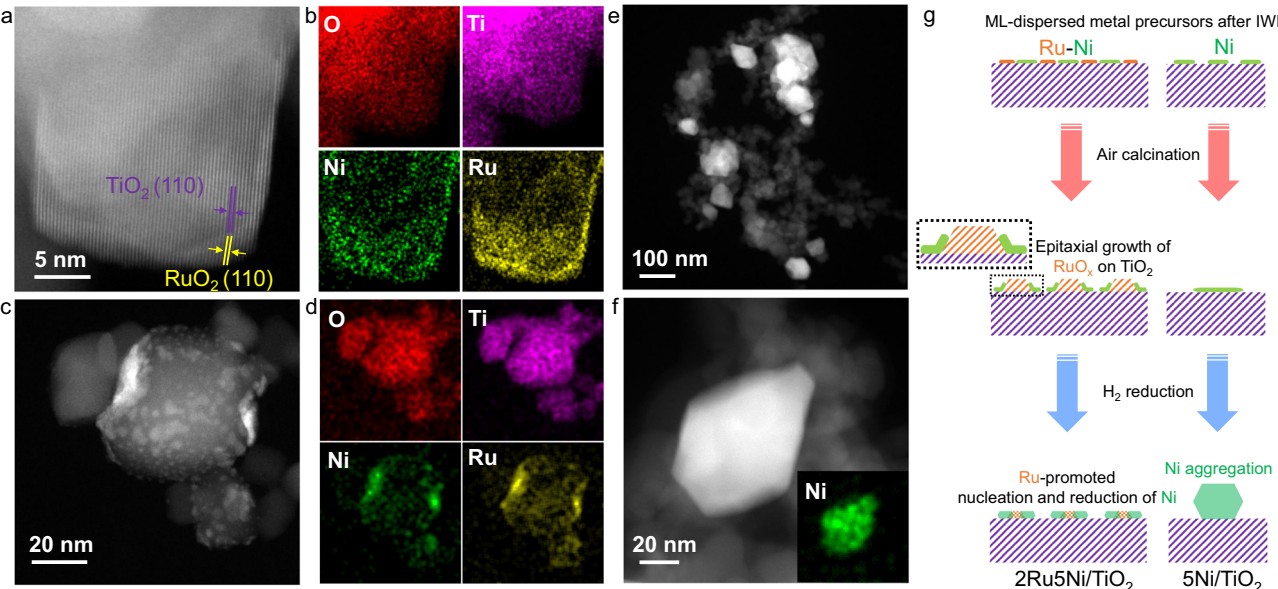

**Fig. 2 | Synthesis and microscopy characterization of catalysts.** HAADF-STEM and corresponding EDS-mapping images of O, Ti, Ni, and Ru elements of 2Ru5Ni/TiO₂ after air calcination (**a**, **b**) and H₂ reduction (**c**, **d**), respectively. **e**, **f** HAADF-STEM images of reduced 5Ni/TiO₂, the inset image in (**f**) corresponds to the EDS-mapping image of Ni element. **g** Schematic illustration of the evolution of bimetallic Ru–Ni and monometallic Ni species on TiO₂ during sequential calcination and hydrogen reduction treatments, respectively.

As shown in Fig. 2c and d, the co-localization of Ru and Ni still existed after reduction, which helps the restricted migration of the total Ni species by Ru species (mean size = 8.1 ± 3.6 nm, Supplementary Figs. 18 and 19). Notably, Ru and Ni species tend to agglomerate slightly during the reaction compared with the STEM maps of spent catalyst (Supplementary Fig. 10). These results indicate that the epitaxial interface between $RuO_2$ and $TiO_2$ leads to the generation of highly dispersed Ru clusters, which further promotes the stability of Ni species due to the strong interaction between Ru and Ni species during the reduction process (Fig. 2g).

The structural properties of different catalysts were further probed by X-ray diffraction (XRD) and X-ray absorption spectroscopy (XAS). As shown in Fig. 3a, the diffraction peaks assigned to metallic Ni ($2\theta = 44.6°$) were sharp for 5Ni/TiO₂, while for 2Ru5Ni/TiO₂ the half-width was much broader, which is consistent with the distinguished Ni distributions in STEM results (Supplementary Figs. 17 and 19). Meanwhile, no obvious peaks of Ru crystal could be detected at $2\theta = 42.2°$, indicating Ru has a great dispersion on the support. In addition, the lower activity of 2Ru8Ni/TiO₂ than 2Ru5Ni/TiO₂ can be explained by the poor Ni dispersion due to the sharper peaks ascribed to metallic Ni observed in Supplementary Fig. 20.

As seen in Fig. 3b, the fitted extended X-ray absorption fine structure (EXAFS) results at Ni K-edge revealed that only Ni–Ni coordination (coordination number of Ni–Ni $(C.N.)_{Ni-Ni} = 8.1$) could be detected in 5Ni/TiO₂, indicating Ni in 5Ni/TiO₂ was close to the state of Ni foil. However, the $(C.N.)_{Ni-Ni}$ of 2Ru5Ni/TiO₂ was only 4.3 (Supplementary Table 3), confirming that Ni species were smaller nanometer-sized clusters. Notably, Ru–Ni scattering could be fitted both at Ru and Ni K-edges for the 2Ru5Ni/TiO₂ sample (Supplementary Figs. 21 and 22), revealing there is interaction between Ru and Ni, which is in agreement with the co-localization of Ru and Ni species as observed in Fig. 2c and d. The much lower coordination number of Ru–Ni scattering $(C.N. = 1.2)$ than Ru–Ru scattering $(C.N. = 4.4)$ demonstrates the Ru atoms still form small clusters instead of dissolving in Ni matrix[33]. This evidence indicates Ru and Ni species are in separated phases with strong interactions rather than a solid-solution alloy[34]. Moreover, compared with 2Ru5Ni/TiO₂, 2Ru/TiO₂ showed a stronger Ru–Ru coordination of $(C.N.)_{Ru-Ru} = 5.8$ (Supplementary Table 4), confirming

the better dispersion of Ru particles in 2Ru5Ni/TiO₂. It can be explained by that Ru–Ni interaction could induce a higher dispersion for Ni and Ru clusters during the reduction process.

The normalized Ru K-edge X-ray absorption near-edge structure (XANES) for 2Ru/TiO₂ and 2Ru5Ni/TiO₂ samples are shown in Fig. 3c. Both 2Ru5Ni/TiO₂ and 2Ru/TiO₂ displayed similar white-line profiles (at ~22138 eV) between $RuO_2$ and Ru foil, suggesting the coexistence of metallic Ru and $RuO_2$ phases of these two samples. Notably, $RuO_2$ still existed after the reduction, demonstrating the epitaxial lattice matching and the strongly trapped Ru species on the TiO₂ support can still exist. Figure 3d presents the XANES spectra at Ni K-edge of 2Ru5Ni/TiO₂, showing higher white-line intensity and blue-shifted edge positions as compared to 5Ni/TiO₂ and Ni foil but distinguished from NiO. It is demonstrated that Ni species in 2Ru5Ni/TiO₂ were close to metallic Ni but part of Ni at interface existed as oxide due to the more dispersed state than 5Ni/TiO₂, which was consistent with the STEM results (Fig. 2c, e).

## Mechanism studies

In situ diffuse reflectance infrared Fourier transform spectroscopy (DRIFTS) analysis was further performed to identify the respective role of Ru and Ni in CO-tolerant toluene hydrogenation reaction (Fig. 4a). First, for 2Ru/TiO₂, after H₂ activation and CO adsorption ($CO_{ad}$), three adsorption bands which related to the linear $CO_{ad}$ on low- (2072 cm⁻¹) and high-coordinated Ru atoms (2009 cm⁻¹), as well as Ru-multicarbonyl species (2133 cm⁻¹) were observed (Fig. 4b: α-1)[35–38]. Then, successively introducing toluene vapor by purging gas to the cell and followed with Ar sweeping brought neglect influence to the DRIFT spectrum (Fig. 4b: α-2). Reversing the above treatments between CO and toluene over another freshly activated 2Ru/TiO₂, interestingly, those characteristic stretching vibration signals of toluene (located at ~2974, ~2935, and ~2878 cm⁻¹) after its adsorption (Fig. 4b: β-1) can be completely replaced by CO (Fig. 4b: β-2)[39,40]. This clearly indicates that CO is more competitive for adsorption than toluene on 2Ru/TiO₂. In contrast, for 2Ru5Ni/TiO₂, red shifts of $CO_{ad}$ from 2079 to 2070 cm⁻¹ and 2016 to 2003 cm⁻¹ were found after successive CO and toluene adsorption (Fig. 4c: α-1, 2), indicating the toluene influence on CO adsorption, since toluene as electron-rich aromatics tends to induce a

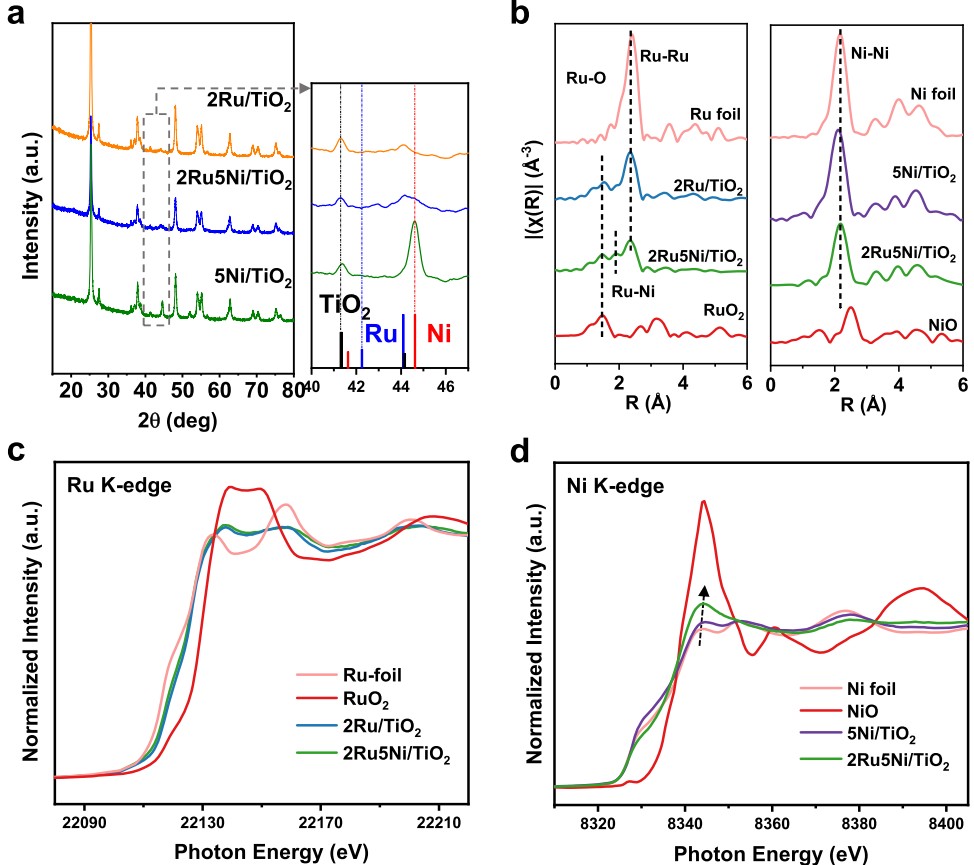

**Fig. 3 | Structural characterization of 2Ru/TiO₂, 2Ru5Ni/TiO₂, and 5Ni/TiO₂ catalysts. a** XRD patterns of the reduced 2Ru/TiO₂, 2Ru5Ni/TiO₂, and 5Ni/TiO₂ with the part of 2θ = 40–47° were enlarged at right. **b** Fourier-transform EXAFS spectra at Ru K-edge (left) and Ni K-edge (right) for 2Ru/TiO₂, 2Ru5Ni/TiO₂, and 5Ni/TiO₂ where Ru, Ni foil, and RuO₂, NiO were used as references. **c** Normalized XANES spectra at Ru K-edge and **d** Ni K-edge for 2Ru/TiO₂, 2Ru5Ni/TiO₂, and 5Ni/TiO₂.

red shift of $CO_{ad}$ if it binds tightly on Ru atoms due to the enhanced electron-back donation from Ru to $CO^{39,40}$. Besides, toluene signals stayed even undergoing CO flowing (Fig. 4c: α-2, β-2), indicating the co-adsorption of toluene and CO on the 2Ru5Ni/TiO₂ surface. Moreover, as seen in Fig. 4c: α-1, only three $CO_{ad}$ bands (2136, 2079, and 2016 cm⁻¹) associated with Ru surface were observed in 2Ru5Ni/TiO₂, and no $CO_{ad}$ bands left in 5Ni/TiO₂ after CO adsorption (Fig. 4d: α-1) which suggests that the interaction between Ni and CO is weak in this condition. In fact, the intensity of absorbed hydrocarbons bands was nearly unchanged even after CO sweeping in Fig. 4d: α-2, β-1, and β-2, confirming CO had a slight influence on the toluene adsorption in 5Ni/TiO₂. The CO-toluene switching results present that there are two different types of active sites on the 2Ru5Ni/TiO₂ for CO and toluene, leading to the non-competitive adsorption between these two reactants. Ru has a stronger affinity to CO which helps the CO activation and methanation, while highly dispersed Ni species act as the main sites for toluene adsorption and hydrogenation in simulated crude hydrogen feed (Fig. 4e). In addition, in situ DRIFT spectra recorded in the reaction condition of CO-tolerant toluene hydrogenation conditions are displayed in Supplementary Fig. 25, further proving the MCH generation in the presence of CO and CO adsorption on Ru species.

To investigate the adsorption properties of RuNiₓ/TiO₂, the hydrogen temperature-programed reduction (H₂-TPR) (Supplementary Fig. 26) and CO temperature-programed desorption (CO-TPD) analysis were conducted (Supplementary Fig. 27). H₂-TPR indicated that the reduction of oxidized Ni species in 2Ru5Ni/TiO₂ (310 °C) was more facile than that of the 5Ni/TiO₂ (360 °C). This should be the consequence of the enhanced hydrogen spillover due to the presence of metallic Ru clusters with lower reduction temperature (i.e., 172 °C,

Supplementary Fig. 26)[41]. The CO-TPD results show a high temperature of CO desorption (522 °C) on 5Ni/TiO₂, which is likely to originate from dissociative adsorption of CO on Ni particles[42,43]. The vanish of this high-temperature CO desorption peak on 2Ru5Ni/TiO₂ further proves the influence of the CO adsorption abilities of Ru on Ni (Supplementary Fig. 27).

## Discussion

Based on all the results above, it can be concluded that there is a strong correlation between the structure modulation of Ru addition and the performance of CO-tolerant hydrogenation reaction for 2Ru5Ni/TiO₂. During the reduction process, the trapped Ru species on TiO₂ kept its high dispersion form, and the Ru–Ni interaction further prevented the agglomeration of Ni, thus promoting the formation of highly dispersed Ni species, which is supposed to be the main active center for CO-tolerant hydrogenation reaction. Instead, the hydrogenation ability for 5Ni/TiO₂ is poor due to its large particle sizes and low dispersion. Besides the structure regulation, DRIFTS results reveal that the non-competitive adsorption of CO and toluene, where CO prefers to be activated on Ru and toluene tends to be hydrogenated on Ni, help the efficient CO-tolerant hydrogenation reaction. Notably, based on XANES results (Fig. 3c-d), the phases of metallic and oxide (interacted with the support) of Ru and Ni both exist. However, from the XPS results in Supplementary Fig. 28, $Ru^0$ and $Ni^0$ are the main sites on the surface of 2Ru5Ni/TiO₂, which means for the surface adsorption and hydrogenation processes, $Ru^0$ and $Ni^0$ play the crucial roles instead of Ru–Ni interface.

In summary, we demonstrate that bimetallic RuNi/TiO₂ is an exceptional catalyst for CO-tolerant toluene hydrogenation. By

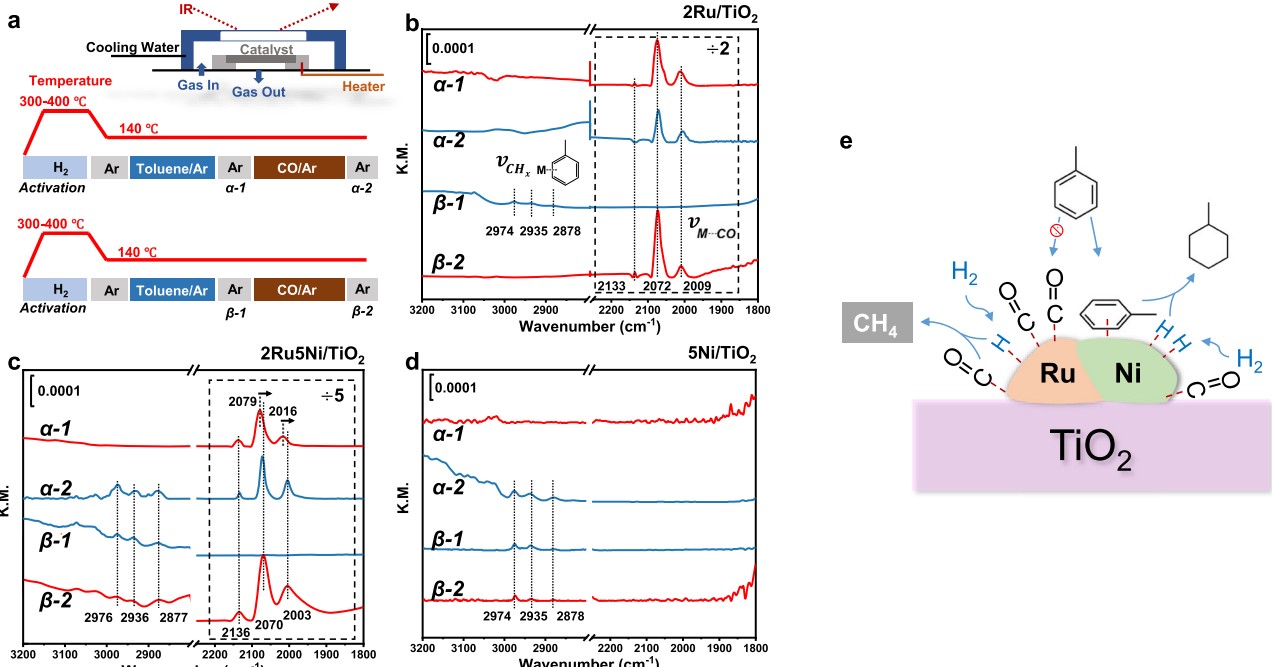

**Fig. 4 | Mechanism studying by in situ DRIFT spectroscopy. a** Diagram of the operations including the pre-reduction and switching details, where α-1, α-2, β-1, and β-2 present the spectra collected in different conditions. α-1 was recorded after CO (CO:Ar = 5:95, v/v) adsorption and Ar sweeping; then followed α-2 recorded for toluene (toluene: Ar = 3:97, v/v) adsorption after CO treatment. After refilling the catalyst, the same process was carried out except toluene adsorption (β-1) was recorded before CO adsorption (β-2). Toluene-CO switching DRIFTS were carried out over **c** 2Ru/TiO$_2$, **b** 2Ru5Ni/TiO$_2$ (the regions of 2300–1800 cm$^{-1}$ were reduced by 2 times for (**c**) and 5 times for (**c**), respectively), and **d** 5Ni/TiO$_2$. Only the regions of wavenumber at 3200–2800 cm$^{-1}$ and 2300–1800 cm$^{-1}$ were shown in the pictures. The velocity of all the carrier gases used in the experiments was 30 mL/min. **e** The mechanism model of RuNi/TiO$_2$ in CO-tolerant toluene hydrogenation reaction.

coupling CO methanation reaction, it dramatically decreases the CO poisoning effect and purifies the crude hydrogen in outlet at atmospheric pressure. The strong interaction between Ru species with the TiO$_2$ support help to inhibit the Ni agglomeration during the reduction process due to Ru–Ni interaction, thus, leading to the formation of highly-dispersed Ni particles. Through in situ DRIFTS, it was found CO was preferentially activated on Ru while toluene adsorption took place on Ni in the presence of CO. The non-competitive adsorption of CO and toluene supports a new route for achieving the goal of crude hydrogen storage, a promising direction to reduce hydrogen energy consumption in the future.

## Methods
### Materials
Analytical grade chemicals including NiCl$_2$·6H$_2$O, RuCl$_3$·xH$_2$O were purchased from Macklin. The commercial TiO$_2$ (P25) was purchased from Degussa AG. Other commercial catalysts including 5Pt/C, 5Pd/C, and 5Ru/Al$_2$O$_3$ were all purchased from Macklin. All chemicals were used as received without any further purification.

### Catalysts preparation
Catalysts with Ru, Ni, and bimetallic RuNi NPs were prepared by incipient wetness impregnation method. In a typical synthesis, for 2Ru5Ni/TiO$_2$ catalyst, 40.8 mg NiCl$_2$·6H$_2$O, and 93 μL RuCl$_3$·xH$_2$O solution (0.43 M) were dissolved in 1.0 mL deionized water. Then 200 mg TiO$_2$ (P25) was wetted by the obtained solution within a 10 mL crucible. Subsequently, the slurry was stirred overnight then dried in a vacuum oven at 60 °C for 12 h and followed with a calcination treatment in air at 300 °C for 2 h. The synthesis procedures for 2Ru/TiO$_2$ and 2RuxNi/TiO$_2$ were the same as 2Ru5Ni/TiO$_2$ except using different amounts of NiCl$_2$·6H$_2$O or RuCl$_3$·xH$_2$O solution. Prior to the catalytic reaction, all calcined samples were reduced in pure hydrogen for 1 h with a heating

rate of 5 °C·min$^{-1}$. The reduction temperature was 300 °C for 2Ru/TiO$_2$, 2RuxNi/TiO$_2$, and 400 °C for 5Ni/TiO$_2$.

### Catalytic evaluation
The catalytic toluene hydrogenation evaluation under pure and crude H$_2$ flow was carried out in a micro-quartz tube reactor (I.D. = 6 mm) at atmospheric pressure (Supplementary Fig. 1). Prior to the catalytic reaction, 0.05 g of catalyst was reduced in a pure H$_2$ (30 mL·min$^{-1}$) at 300 or 400 °C for 1 h. After reduction and cooling the reactor to room temperature, toluene vapor was introduced to catalyst bed through a bubbling flask (keep the temperature by water bath at 7 °C and P$_{toluene}$ = 0.014 bar or 19.2 °C and P$_{toluene}$ = 0.028 bar) with either pure or crude H$_2$ flow (30 mL·min$^{-1}$) at programmed temperature. The gas hourly space velocity (GHSV) of carrier gas was maintained at 36,000 mL g$_{cat}^{-1}$ h$^{-1}$ with WHSV (weight hourly space velocity) of toluene = 2.1 h$^{-1}$ for a typical temperature-dependent test. For CO-tolerant ability and stability test, crude hydrogen containing various CO concentration of 0.1–1% (CO/Ar/H$_2$ = 0.1/0.1/99.8, CO/Ar/H$_2$ = 0.5/0.5/99, CO/Ar/H$_2$ = 1/1/98, v/v) was included at a rate of 10 mL·min$^{-1}$, where the temperature of water bath was kept at 19.2 °C (P$_{toluene}$ = 0.028 bar), and GHSV of the carrier gas was maintained at 12,000 mL g$_{cat}^{-1}$ h$^{-1}$ with WHSV of toluene = 1.4 h$^{-1}$. The composition of outlet gases was analyzed online using a GC-7890B gas chromatograph with a Porapak Q plot column before TCD to detect the H$_2$, Ar, CH$_4$, CO, and a FID equipped with Innowax capillary column to detect the hydrocarbons such as CH$_4$, toluene and methyl cyclohexane.

The total conversion of CO was calculated according to the following equation:

$$X_{co} = \frac{[CO]_{in} - [CO]_{out}}{[CO]_{in}} \qquad (1)$$

Calculation of the MCH yield was based on the following equation:

$$\text{Yield}_{\text{MCH}} = \frac{S_{\text{MCH}}}{S_{\text{Toluene}} + S_{\text{MCH}}} \times 100\% \quad (2)$$

$S_{\text{MCH}}$ and $S_{\text{Toluene}}$ represent the integral area of MCH and toluene detected from FID in gas chromatograph. Noted the chromatographic correction factor is nearly the same for toluene and MCH. Since MCH selectivity > 99% among the products (Supplementary Fig. 3), other byproducts such as methyl-cyclohexene is not included in the yield calculation. And methane was also the main product with selectivity > 95% from CO hydrogenation (Supplementary Fig. 3).

Calculation of toluene conversion rate was based on the following equation:

$$\text{Rate}(\text{mol}_{\text{toluene}}/\text{mol}_{\text{metal}}/h) = \frac{\text{WHSV}_{\text{toluene}}/M_{\text{toluene}}}{\text{Mole}_{\text{metal}}} \times \text{conversion(toluene)} \quad (3)$$

$P_{\text{toluene}}$ was calculated according to the Antoine equation:

$$\log(P_{\text{toluene}}) = A - \left(\frac{B}{T+C}\right) \quad (4)$$

T represents the temperature of the toluene; A, B, C are constants of 4.24, 1426.45, −45.96 at 273–298 K, respectively[44].

## Structural characterization

**XRD analysis.** The measurements were performed on a Rigaku D/MAX-PC 2500 diffractometer using Cu Kα radiation ($\lambda = 0.15418$ nm) at 40 kV and 100 mA with a scanning rate of $10° \text{ min}^{-1}$ and a $2\theta$ angle ranging from 5° to 80°.

**ICP-AES analysis.** ICP-AES measurements were carried out on a Prodigy 7 Inductively Coupled Plasma-Atomic Emission Spectrometer (Leeman Ltd.). The samples (-10 mg) were digested with 0.75 mL hydrochloric acid, 0.25 mL nitric acid, and 0.1 mL hydrofluoric acid at 260 °C in an UltraWAVE ECR Microwave Digestion System (Milestone Ltd.).

**XAFS analysis.** The X-ray absorption fine structures (XAFS) measurement at Ru K-edge and Ni K-edge were carried out at the BL14W beamline in Shanghai Synchrotron Radiation Facility (SSRF, Energy 3.5 GeV, Current 250 mA in maximum). The XAFS spectra were collected in fluorescence mode, using the Lytle detector. Ru foil, $RuO_2$, and Ni foil, NiO (Adamas) were used as standards. Before measurement, all samples were pre-reduced and transferred into the glove box for pelleting and sealing (the reduction temperature is consistent with that used for catalytic tests). All XAS spectra were analyzed using the Ifeffit package version 1.2.11.

**XPS analysis.** The XPS spectra of the activated samples were collected using an Axis Ultra Imaging Photoelectron Spectrometer (Kratos Analytical). The activated samples were transferred from glove box directly into the ultrahigh vacuum chamber without exposure to air for XPS measurement at room temperature without exposure to air. The XPS spectra were processed using CasaXPS software.

**Electron microscopy.** Aberration-corrected scanning transmission electron microscopy (AC-STEM) was operated on a ThermoFisher Themis Z transmission electron microscope equipped with two aberration correctors, operated at 300 kV with a convergence semi-angle of 17.8 or 25 mrad. Energy-dispersive X-ray spectroscopy (EDS) was carried out using 4 in-column Super-X detectors. Integrated differential phase contrast (iDPC)-STEM were recorded using a camera length of 285 mm on DF4 detectors, inner- and outer collection angles of 5 and 26 mrad, respectively. High angle annular dark-field (HAADF)-STEM images were recorded using a camera length of 115 mm on HAADF detectors, and inner- and outer collection angles of 47 and 200 mrad, respectively. The acquired images were processed by Gaussian blur filter to reduce the low-frequency information.

**iDPC-assisted HAADF-STEM imaging method.** iDPC-STEM imaging was first used for searching the region of interest and imaging adjustments. $TiO_2$ materials were able to be visualized at atomic-resolution under iDPC-STEM imaging mode at an ultralow bean current smaller than 1.0 pA that has minimal electron beam effect. Once the samples were in focus under iDPC-STEM mode, the microscope was switched to HAADF-STEM mode to take only one shot at a beam current of only 30.0–100.0 pA. Ru and Ni species are easy to be identified in HAADF-STEM images based on Z-contrast. A vacuum transfer holder (Fischione Model 2550) was used to eliminate additional air exposure that might change the valence states of Ni.

**$H_2$ Temperature programmed reduction analysis.** First, 50 mg of sample was filled in a fixed-bed reactor. After sweeping in Ar at 120 °C for 1 h to remove the residual water, the reactor was then cooled to room temperature. Subsequently, the temperature was increased to 650 °C at the rate of 10 °C/min in 30 mL/min of 10%$H_2$/Ar with the intensity of $H_2$ ($m/z = 2$) was monitored by mass spectrometry (MS).

**CO programmed desorption analysis.** First, 50 mg of sample was filled in a fixed-bed reactor. After reduction process, the feed was switched from $H_2$ to Ar then cooled to room temperature. For CO-TPD analysis, 10%CO/Ar was introduced for 1 h to reach the CO saturated adsorption. Then pure Ar was used to remove the gas and weakly adsorbed CO. Subsequently, the temperature was increased to 650 °C at the rate of 10 °C/min in 30 mL/min of Ar with the intensity of CO ($m/z = 28$) was monitored by mass spectrometry (MS).

**DRIFTS analysis.** The in situ diffuse reflectance infrared Fourier transform spectra (DRIFTS) of different catalyst were performed in a reaction cell on a VERTEX 70 spectrometer equipped with a liquid nitrogen cooled mercury–cadmium–telluride (MCT) detector and an in situ Praying Mantis diffuse reflection reaction cell (Harrick Scientific). In a typical measurement, first, the catalyst was diluted with BN and packed smoothly in the tank within the reaction cell. Afterward, the catalyst was reduced at 300 °C in pure hydrogen (30 mL/min) for 1 h then switched feed to Ar to sweep out the $H_2$. After the temperature cooled to 140 °C, the baseline was collected and the gas flow was switched to the CO/Ar gas (CO/Ar = 5/95, v/v) for 30 min. Then Ar (30 mL/min) was switched into the cell again and collected the spectra until reaching balance (α-1). Subsequently, a four-way valve was switched to bubble the toluene steam (temperature of water bath was kept at 20 °C) into the cell by Ar for 30 min then switched the four-way valve to bypass the toluene. After the spectra became steady, they were collected named α-2. Then the cell was refilled with fresh catalyst followed with reduction procedure. *β*-related spectra were collected the same way as *α* except toluene was introduced before CO.

## Data availability
The data that support the plots within this paper and other finding of this study are available from the corresponding author upon reasonable request. Source data are provided as a Source data file. Source data are provided with this paper.

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

## Acknowledgements

This work was financially supported by the National Key R&D Program of China (2021YFA1501102) and the National Natural Science Foundation of China (21725301, 21932002, 21821004). S.D. thanks the Fundamental Research Funds for the Central Universities, Shanghai Rising-star Program (20QA1402400), and the Program for Professor of Special Appointment (Eastern Scholar) at Shanghai Institutions of Higher Learning. Additional support was provided by the Frontiers Science Center for Materiobiology and Dynamic Chemistry and the Feringa Nobel Prize Scientist Joint Research Center. D.M. acknowledges support from the Tencent Foundation through the EXPLORER PRIZE.

## Author contributions

D.M. conceived the project. D.M. and S.D. supervised the study. Z.W. and C.D. performed most of the reactions. X.T. and S.D. performed the electron microscopy study. C.D., M.P., and X.Q. performed the X-ray structure characterization (XAS and XRD) and analysis. Z.W. and J.Z. did the DRIFS experiments. Y.X. and C.S. did the XPS experiments. Z.W., Xuan Liang, and Xingwu Liu performed the substrates extension experiments. Z.W., C.D., X.T., S.D., and D.M. wrote the paper. All the authors discussed and revised the paper.

## Competing interests

The authors declare no competing interests.
