## [Peer Review File · Nature Communications]

CO-tolerant RuNi/TiO₂ Catalyst for Crude Hydrogen Storage and PurificationREVIEWER COMMENTS

Reviewer #1 (Remarks to the Author):

This paper reports an interesting investigation of the simultaneous hydrogenation of CO (a poison) and toluene (a H-storage molecule). The topic is relevant to H-storage in liquid organic hydrogen carrier (LOHC). The paper is globally well structured and the results reported support the proposed conclusions that Ni-Ru/TiO₂ are promising catalysts. There are yet two important aspects to consider in more detail.

1- The temperatures used are quite low and CO may readily react with metallic Ni to form volatile and highly toxic Ni(CO)₄. This may result in catalyst corrosion and Ni redistribution throughout the reactor and setup. This is discussed in detail by Meunier et al. (Journal of Catalysis 372 (2019) 388) and the references therein. The authors should prove that no Ni(CO)₄ is formed under the reaction conditions used (preferably by monitoring reactor exhaust by mass spectrometry or transmission IR).
2- The authors explained the loss of activity of Ru/TiO₂ at higher temperatures by a loss of toluene adsorption. Yet, H₂ adsorption could also become limited at higher temperatures and may be the main origin of activity loss. This should be discussed.

Reviewer #2 (Remarks to the Author):

The paper by Wang et al. tackles a combination of very interesting and hot subjects, namely CO-tolerant catalyst development for crude hydrogen storage via the LOHC technology. This reviewer ranks the experimental catalysis, microscopic and spectroscopic work in this paper as groundbreaking, however, the conclusion drawn from the CO-tolerant hydrogenation studies as well as its reaction mechanism need further improvement. This is why, this reviewer ranks the paper as 'major revision' but would strongly support publication in Nature Communications after addressing the open points in the results analysis section in an appropriate manner:

Firstly, there are some confusions:

1. There is some confusion with the "Mild Conditions" definition in the title of this manuscript. Toluene hydrogenation can be easily hydrogenated with pure H₂ at reaction temperature below 100 °C. (ACS Omega 2021, 6, 8, 5846–5855: Under 30 °C, atmospheric H₂ pressure, and a toluene/(Pt+Rh) molar ratio of 200/1, Pt_{0.77}Rh₁ alloy catalyst reach 100.0% of methyl cyclohexane). The definition of "mild conditions" should be consistent and correct throughout this paper with previous studies.

2. The statement in lines 57-59 is confusing too (However, as the benzene ring acts as the main unsaturated hydrogen storage unit, the essential ensemble-requirement of benzene hydrogenation would inevitably inhibit the application of single-atom catalyst in CO-tolerant hydrogen storage. 15-19). The authors cite references here that deal mostly with metal particles catalyst (for which the statement is more correct). Single atom has been demonstrated to hydrogenate toluene with a higher reaction rate than particle catalyst. (Nat Commun 13, 1092 (2022). <https://doi.org/10.1038/s41467-022-28607-y>;) Single atom catalyst has not been tested for toluene hydrogenation in the CO-tolerant case, but it has been claimed that a single atom (Pt) catalyst having a lower binding energy with CO than Pt cluster/nanoparticles. (H.V. Thang et al. / Journal of Catalysis 367 (2018) 104–114 107), so this reviewer suggests the authors make a clear discussion on this topic.

Secondly, about the reaction pathway and mechanism. The authors provided a beautiful study on microscopic and spectroscopic side and experimental catalysis side. But only an FT-IR is not efficient to bridge them up for reaction pathway and mechanism discussion.

1. The author should show the results of selectivity. Are there any by-products such as methyl cyclohexene or -diene? The author should make a discussion on that.

2. In heterogeneous catalysis, directly bridge reactant adsorption (FT-IR in this case) with reaction activity is not efficient. In some cases, suitable adsorption strength, as well as pre-activation is important. Some stronger adsorptions may cause the poison effect. The authors should make a clear

discussion on the specific case of toluene hydrogenation (in CO). (This study is a great case to talk about the adsorption-activation-reaction process and competing adsorption. and a clear discussion will strength this study to the level of Nature Communications.)

3. To further clarify the adsorption issue, this reviewer suggests that a toluene and CO Temperature-programmed desorption on the three catalysts sample should be provide and discussed.

4. The authors provided a lot of characterization data on catalytic materials side. This reviewer suggests the authors to combine these data with catalysis to propose a reaction mechanism model. Such as combine the metal-support interaction, and metal (Ru)-metal (Ni) interactions to further explain their distinct binding properties. And provide more insight for the conclusion of "Ru and Ni served as the active sites for CO methanation and toluene hydrogenation respectively."

5. Is the interface of Ru-Ni matter for the reaction?

Reviewer #3 (Remarks to the Author):

The RuNi cluster on TiO₂ wad design for stable activity for hydrogenation even with CO impurity. This paper can be accepted after considering followings.

1) Ru and Ni can form solid-solution alloy or separated cluster?

2) The electronic state of Ru and Ni can be modified by the alloying or clustering?

3)Why TiO₂ is suitable support for the formation of this alloy cluster.

4)The reference paper on RuNi/TiO₂ reported at ChemCatChem 2018, 10, 3526-3531 should be cited and compared.

Reviewer #4 (Remarks to the Author):

The manuscript describes the development of bimetallic RuNi@TiO₂ catalysts with the anti-CO-positioning ability for potential application in a LOHC-based hydrogen storage system, in detail the toluene-MCH pair and the concept was demonstrated in the hydrogenation step of toluene hydrogenation to MCH for H₂ storage. This is an important topic for investigation and the findings from the current work do show the progress beyond the state of the art and provided some insights into the relevant mechanisms of the system under investigation. I am not the expert in all the areas included in the current work, but after reading it carefully, I identified the following comments/queries for the authors to consider for revision and improvement.

1. The title claimed mild conditions, and I think this should be reflected by a revised abstract and conclusion, i.e. < 200C at atmospheric pressure, indeed being much lower than the common conditions of 200-300 C and 10-50 bar.

2. Following the comment above, I think the authors need to explain why such high activity was achieved by the current catalytic system at quite short residence time since hydrogenation reactions normally requires elevated P and T.

3. The current catalytic systems using a bubbler for vapour phase reaction, being rather dilute. Can the authors try the conventional packed/trickle bed reactors or batch reactor with gas-liquid systems to show that the developed catalyst is not specific for the microreactor with gas phase reactions?

4. Exemplar GC analysis of products from the rig is needed in the revised SI.

5. Deactivation regarding MCH yield was measured during the longevity test, which was not discussed, though comment stating it stayed over 60% was provided. The authors need to elucidate this aspect since maintaining MCH yield is the key target of the work, yet continuous decline of the yield is very obvious.

6. Regarding the discussion for CO₂ methanation, I think the authors oversell the performance. CO₂

hydrogenation requires much higher temperatures than the current 100-220C, therefore, a simple discussion will be fine rather than being surprising about the phenomenon of non CO₂ conversion in the current system.

7. Generic feature of the develop catalyst is needed, i.e. for hydrogenation of other LOHC pairs such as DBT to PDBT and/or benzene to cyclohexane.

Response to reviewers

We thank the reviewers for the very positive views and constructive comments. We have addressed all the comments point-by-point and revised the manuscript accordingly. In this response letter, comments from the reviewers are summarized with the major concerns quoted in blue italic.

Reviewer #1

This paper reports an interesting investigation of the simultaneous hydrogenation of CO (a poison) and toluene (a H-storage molecule). The topic is relevant to H-storage in liquid organic hydrogen carrier (LOHC). The paper is globally well structured and the results reported support the proposed conclusions that Ni-Ru/TiO₂ are promising catalysts. *There are yet two important aspects to consider in more detail.*

Response to comments: We thank the reviewer for the very positive view of our work. Per the request of the reviewer, we have added additional discussion in the following point-by-point response and the revised manuscript.

1-The temperatures used are quite low and CO may readily react with metallic Ni to form volatile and highly toxic Ni(CO)₄. This may result in catalyst corrosion and Ni redistribution throughout the reactor and setup. This is discussed in detailed by Meunier et al. (Journal of Catalysis 372 (2019) 388) and the references in there. The authors should prove that no Ni(CO)₄ is formed under the reaction conditions used (preferably by monitoring reactor exhaust by mass spectrometry or transmission IR).

Response to comments: We appreciate the reviewer's comment. As we know, the Ni(CO)₄ will cause the Ni leaching from the catalyst. To clarify whether Ni(CO)₄ is formed during the reaction, we compared the Ni contents of 2Ru5Ni/TiO₂ catalysts before and after reaction by ICP-AES, and found the contents are almost the same (**Supplementary Table 1** or **Table R1**). This evidence eliminates the leach of Ni. In addition, in **Figure 4c** (or **Figure R1**), the CO adsorption bands observed in DRIFTS results at 2136, 2079 and 2016 cm⁻¹ could not match with the Ni(CO)₄ specific band at 2055 cm⁻¹ (*J. Catal.* **2019**, 372, 388), indicating Ni(CO)₄ is not likely to generate during the reaction.

Table R1. Ni contents from ICP-AES results of fresh and used 2Ru5Ni/TiO₂ catalysts.

Sample	Ni content (%)
2Ru5Ni/TiO ₂ -fresh	4.72
2Ru5Ni/TiO ₂ -spent ^a	4.75

^a The spent samples were taken after 24 h continuous reaction. Reaction conditions: 180 °C, 0.5 %CO/H₂, GHSV of the carrier gas = 12,000 mL/g_{cat}/h and WHSV of toluene = 1.4 h⁻¹.

Figure R1 Toluene-CO switching DRIFTS on 2Ru5Ni/TiO₂.

2-The authors explained the loss of activity of Ru/TiO₂ at higher temperatures by a loss of toluene adsorption. Yet, H₂ adsorption could also become limited at higher temperatures and may be the main origin of activity loss. This should be discussed.

Response to comments: We thank this reviewer for the insightful comment. On Ni/Al₂O₃, Salmi and Smeds found the hydrogenation of toluene had the highest reaction rate at appropriate 170 °C and ascribed it to the escape of catalytically active hydrogen from the Ni-surface at the highest reaction temperatures. (*Chemical Engineering Science*, **1993**, 48, 3813-3828). Meanwhile, Keane also observed the same trend of Ni/SiO₂ in xylene hydrogenation that the TOF exhibited a maximum at 175 °C. However, they ascribed it to a critical loss of the reactive aromatic species from the surface. (*J. Catal.* **1997**, 166, 347–355) So in different systems, it's likely either H₂ or toluene adsorption could be the reason that inhibiting the aromatics hydrogenation reaction at a higher temperature. Therefore, we agree with the reviewer's comment, and we have revised the explanation of "The significant activity loss of

2Ru/TiO₂ with the temperature increasing from 120 °C to 200 °C can be ascribed to the continuously enhanced toluene desorption” to “The significant activity loss of 2Ru/TiO₂ with the temperature increasing from 120 °C to 200 °C can be ascribed to the continuously enhanced toluene or hydrogen desorption”. The two references have also been cited in the manuscript.

Reviewer #2

The paper by Wang et al. tackles a combination of very interesting and hot subjects, namely CO-tolerant Catalyst development for Crude Hydrogen Storage via the LOHC technology. This reviewer ranks the experimental catalysis, microscopic and spectroscopic work in this paper as ground-breaking, *however, the conclusion drawn from the CO-tolerant hydrogenation studies as well as its reaction mechanism need further improvement*. This is why, this reviewer ranks the paper as major revision' but would strongly support publication in Nature communications after addressing the open points in the results analysis section in an appropriate manner.

Response to comments: We thank the reviewer for the very positive view of our work. Per the request of the reviewer, we have added additional experimental results and discussion in the following point-by-point response and the revised manuscript.

Firstly, there are some confusions:

1. there is some confusion with the “Mild Conditions” definition in the title of this manuscript. Toluene hydrogenation can be easily hydrogenated with pure H₂ at reaction temperature below 100 °C. (ACS Omega 2021, 6, 8, 5846–5855: Under 30 °C, atmospheric H₂ pressure, and a toluene/(Pt+Rh) molar ratio of 200/1, Pt0.77Rh1 alloy catalyst reach 100.0% of methyl cyclohexane). The definition of “mild conditions” should be consistent and correct throughout this paper with previous studies.

Response to comments: We thank the reviewer for pointing out this mistake. Indeed, the expression “mild” is not that suitable here as toluene hydrogenation (with pure H₂) can take place at room temperature over some supported metal catalysts (*J. Am. Chem. Soc.* **2010**, 132, 6541-6549 and reviewers' citations mentioned). Therefore, we revised the title to “CO-tolerant RuNi/TiO₂ Catalyst for Crude Hydrogen Storage and Purification”. In addition, we have revised the expression of “mild condition” in the manuscript to practical experimental conditions.

2. the statement in lines 57-59 is confusion too (However, as the benzene ring acts as the main unsaturated hydrogen storage unit, the essential ensemble-requirement of benzene hydrogenation would inevitably inhibit the application of single-atom catalyst in CO-tolerant hydrogen storage.¹⁵⁻¹⁹). The authors cite references here that deal mostly with metal particles catalyst (for which the statement is more correct). Single atom has been demonstrated to hydrogenate toluene with a higher reaction rate than particle catalyst. (Nat Commun 13, 1092 (2022), <https://doi.org/10.1038/s41467-022-28607-y>;) Single atom catalyst has not been tested for toluene hydrogenation in the CO-tolerant case, but it has been claimed that a single atom (Pt) catalyst having a lower binding energy with CO than Pt cluster/nanoparticles. (H.V. Thang et al. / Journal of Catalysis 367 (2018) 104–114 107), so this reviewer suggests the authors make a clear discussion on this topic.

Response to comments: We thank the reviewer for the comment. Based on the reference provided by the reviewer, we admit that under certain circumstance, the hydrogenation of aromatic substrate which generally requires metal ensemble sites can take place favorably at Pt single atom sites. In addition, it is also widely accepted that with less metallic character, metal single atoms tend to bind CO in a less intensive manner. Despite the theoretical potential of single atom catalysts (SACs) in driving the CO-tolerant toluene hydrogenation process, however, we believe the attenuated interaction between SACs and CO would inevitably, compromise the purification of the crude hydrogen feed. Alternatively, the parallel CO methanation along with toluene hydrogenation offers a more practical solution to alleviate the CO-poisoning effect of the metal to achieve simultaneous crude hydrogen storage and purification.

According to the above discussion, the second paragraph of the **Introduction** has been revised accordingly. We also cited those important references that the reviewer mentioned.

Secondly, about the reaction pathway and mechanism. The authors provided a beautiful study on microscopic and spectroscopic side and experimental catalysis side. But only a FT-IR is not efficient to bridging them up for reaction pathway and mechanism discussion.

1. The author should show the results of selectivity. Is there any by products such as methyl cyclohexene or -diene? The author should make a discussion on that.

Response to comments: We thank the reviewer for this suggestion. During the reaction process of 2Ru5Ni/TiO₂, with toluene/CO/H₂ as the inlet reactant, MCH was detected as the dominant toluene

hydrogenation product (**Supplementary Figure 3** or **Figure R2**). The only detectable byproduct is cyclohexene, with less than 0.2% selectivity, which can be regarded as trace amount.

Figure R2. Typical GC analysis graph of CO-tolerant toluene hydrogenation using 2Ru5Ni/TiO₂.

2. In heterogeneous catalysis, directly bridge reactant adsorption (FT-IR in this case) with reaction activity is not efficient. In some cases, suitable adsorption strength, as well as a pre-activation is

important. Some stronger adsorptions may cause the poison effect. The authors should make a clear discussion on the specific case of toluene hydrogenation (in CO). (This study is a great case to talk about the adsorption-activation-reaction process and competing adsorption and a clear discussion will strength this study to the level of Nature Communications.)

Response to comments: We thank the reviewer for this insightful comment. To discuss the poison effect of CO on 2Ru5Ni/TiO₂, we compare the catalytic hydrogenation performance of toluene in pure hydrogen and crude hydrogen (**Figure 1a** or **Figure R3**), while 0.1 % CO could induce the lower MCH yield especially for 2Ru/TiO₂ and commercial catalysts (“However, along with the addition of 1000 ppm CO...demonstrating the feasibility of crude hydrogen storage” in manuscript). Based on these results, we think the strongest poison species was CO in this reaction. For the pre-activation process, we also believe it is very important and can influence the catalytic performance greatly. So, during our DRIFTS experiments, we kept the pre-activation processes the same as real conditions. Although we think our DRIFTS results could elucidate the competing adsorption relationship of CO and toluene on the catalysts, we agree with the reviewer’s opinion “directly bridging reactant adsorption with reaction activity is not efficient”, and there are differences between the reactant adsorption and the reaction in true conditions. So, we added the *in-situ* DRIFTS spectra in CO-tolerant toluene hydrogenation conditions (**Supplementary Figure 25** or **Figure R4**), and the result shows that the MCH could generate in the presence of CO and CO adsorption on Ru species were also observed. More discussions were replenished in Supplementary Note 2 and “In addition, the results of *in-situ* DRIFTS ...and CO adsorption on Ru species.” marked in blue in the manuscript.

Figure R3. The metal-normalized toluene conversion rates at 170 °C over different catalysts using either pure hydrogen or 0.1 %CO/H₂ (CO:Ar:H₂ = 0.1:0.1:99.8, v/v, GHSV (gas hourly space velocity) = 36,000 mL/g_{cat}/h) as carrier gases. The conversion rates of toluene were measured in kinetic region (i.e., < 15 %).

Figure R4. In situ DRIFTS of CO-tolerant hydrogenation reaction over 2Ru5Ni/TiO₂ catalyst. The spectrum of 0 min represents the balance state of toluene hydrogenation under pure H₂ in the presence of 0.05 bar toluene. Then the carrier gas was switched to 0.5 %CO/H₂, and the spectra from 0 min to 60 min were recorded. Test conditions: all the velocity of the feed= 20 mL/min, the temperature of the cell was kept at 180 °C.

3. To further clarify the adsorption issue, this reviewer suggests that a toluene and CO Temperature-programmed desorption on the three catalysts sample should be provide and discussed.

Response to comments: We appreciate this reviewer’s comment. The CO-TPD on three catalysts samples has been added in the **Supplementary Figure 27** (or **Figure R5**) and related discussion was added in Supplementary Note 4 and “The CO-TPD results show ...the CO adsorption abilities of Ru on Ni” marked in blue in the manuscript.

Figure R5. CO-TPD patterns of 2Ru/TiO₂, 2Ru5Ni/TiO₂ and 5Ni/TiO₂.

Toluene-TPD we carried out but no toluene desorption was observed from 50 – 600 °C. From the **Figure R6a** as provided in the following, the toluene represented by MS signal of m/z=92 kept stable without peaks for the three samples. However, we found the desorption of H₂, C₂H₄ (signals of m/z=2 and m/z=28, **Figure R6b**), and the same trends of these two signals indicate that adsorbed toluene cracked to hydrogen, lower hydrocarbon and carbon deposition on the surface in the TPD conditions (Notably, in the real reaction conditions, the lower temperature and high partial pressure of H₂ can inhibit this process). That is why we didn’t observe the desorption of signal of m/z=92 and the date was not displayed in the **SI**.

Figure R6. Toluene-TPD results of the reduced 2Ru/TiO₂, 2Ru5Ni/TiO₂ and 5Ni/TiO₂.

4. The authors provided a lot of characterization data on catalytic materials side. This reviewer suggests the authors to combine these data with catalysis to propose a reaction mechanism model. Such as combine the metal-support interaction, and metal (Ru)-metal (Ni) interactions to further explain their distinct binding properties. And provide more insight for the conclusion of “Ru and Ni served as the active sites for CO methanation and toluene hydrogenation respectively.”

Response to comments: We thank for the reviewer for the insightful comment. The mechanism model (the following picture) has been added as **Figure 4e** (or **Figure R7**) in the revised manuscript. Deeper discussion on the roles of Ru and Ni in the reaction was revised as “To investigate the adsorption properties ... the crucial role instead of Ru-Ni interface” in the manuscript.

Figure R7. The mechanism model of RuNi/TiO₂ in CO-tolerant toluene hydrogenation reaction.

5. Is the interface of Ru-Ni matter for the reaction?

Response to comments: We appreciate this reviewer’s comment, and it is an important question on the roles of Ru and Ni during reaction. From the related characterization results, we have drawn the conclusion that Ru has a stronger affinity to CO to help the CO activation and methanation, while Ni species, which are highly dispersed with the help of Ru, act as the main sites for toluene adsorption and hydrogenation in simulated hydrogen feed. As for the Ru-Ni interface, we think it is not the key point in this reaction. The reason is, as XANES results show (**Figure 3c, d** or **Figures R8a, b**), the phases of metallic and oxide of Ru and Ni (interacted with the support) both exist. However, from the added XPS results (**Supplementary Figure 28** or **Figure R9**), Ru⁰ and Ni⁰ are the main sites on the surface of 2Ru5Ni/TiO₂, which means for the surface adsorption and hydrogenation processes, Ru⁰ and Ni⁰ play the crucial roles instead of Ru-Ni interface. The revised discussion was provided as “Notably, though from XANES results...the crucial role instead of Ru-Ni interface” marked in blue in the manuscript.

Figure R8. (a) Normalized XANES spectra at Ru K-edge and (b) Ni K-edge for 2Ru/TiO₂, 2Ru5Ni/TiO₂ and 5Ni/TiO₂.

Figure R9. XPS spectra of the reduced 2Ru/TiO₂, 2Ru5Ni/TiO₂ and 5Ni/TiO₂.

Reviewer #3

The RuNi cluster on TiO₂ was designed for stable activity for hydrogenation even with CO impurity. This paper can be accepted after considering the followings.

Response to comments: We thank the reviewer for the very positive view of our work. Per the request of the reviewer, we have added additional experimental results and discussion in the following point-by-point response and the revised manuscript.

1) Ru and Ni can form solid-solution alloy or separated cluster?

Response to comments: We thank the comment from the reviewer, and this is a crucial question for the structure of our catalysts. Based on our experimental evidence, the Ru and Ni species are in separated phases with strong interactions rather than a solid-solution alloy due to the main reasons as follows.

1. The content of Ru (2 wt.%) is lower than that of Ni (5 wt.%) in 2Ru5Ni/TiO₂ catalyst. If the RuNi is a solid-solution alloy, the scattering of Ru-Ni in FT-EXAFS should be dominant since Ru atoms distribute uniformly and bond with more hetero-Ni atoms (*Nano Lett.* **2021**, 21, 9293-9300). However, according to FT-EXAFS result for 2Ru5Ni/TiO₂, the coordination number of Ru-Ni scattering (CN=1.2) of Ru K-edge is much lower than that of Ru-Ru scattering (CN=4.4), as shown

in **Supplementary Table 4** (or **Table R2**) in the supporting information. This evidence demonstrates that the Ru atoms still agglomerate as small clusters instead of solving in Ni matrix.

Table R2. The fitting results of the EXAFS spectra of Ru K-edge for 2Ru/TiO₂ and 2Ru5Ni/TiO₂ with different operations.

Sample	Shell	C.N. ^a	Bond length(Å)	ΔE_0 (eV)	σ^2 (Å ²) ^b	R-factor
2Ru/TiO ₂	Ru-Ru	5.8	2.67	-4.9	0.004	0.003
	Ru-O	3.1	1.98		0.010	
	Ru-Ru	4.4	2.66		0.005	
2Ru5Ni/TiO ₂	Ru-O	4.4	2.01	-3.2	0.015	0.001
	Ru-Ni	1.2	2.55		0.007	
	Ru-Ru	5.1	2.65		0.004	
2Ru5Ni/TiO ₂ -used	Ru-O	2.3	1.96	-6.5	0.012	0.01
	Ru-Ni	1.4	2.58		0.001	

^a Coordination number.

^b Mean squared displacement.

- Based on the FT-EXAFS of Ru K-edge for 2Ru5Ni/TiO₂ after reduction, the Ru-O scattering still exists (CN=4.4), indicating that part of RuO₂ is stabilized on TiO₂ instead of forming RuNi alloy since relatively high metal-O scattering should not be observed in bimetallic alloys. The evidence of Ru-O scattering also demonstrates the separated phases of Ru- and Ni-based species.

To clarify this point, the previous description “NiRu cluster” is now changed to “Ni-Ru cluster” to highlight the separated phases.

2) The electronic state of Ru and Ni can be modified by the alloying or clustering?

Response to comments: We thank this reviewer for the insightful comment. According to the XANES result, the oxidation states of Ru are similar in 2Ru5Ni/TiO₂ and 2Ru/TiO₂ (**Figure 3c** or **Figure R8a**). However, in the Ni K-edge of XANES (**Figure 3d** or **Figure R8b**), these two catalysts possess different oxidation states of Ni species in 2Ru5Ni/TiO₂ and 5Ni/TiO₂. As for 5Ni/TiO₂, Ni is in a metallic state. However, Ni shows oxidation evidence in 2Ru5Ni/TiO₂ which should be attributed to the interaction with the adjacent Ru.

3) Why TiO₂ is suitable support for the formation of this alloy cluster.

Response to comments: We appreciate the reviewer's comment. The TiO₂ support is important in stabilizing the RuO₂ in 2Ru5Ni/TiO₂ catalyst during calcination process. The d-spacings of rutile TiO₂(110) (3.28Å) and RuO₂(110) (3.23Å) are quite close, and this introduces an epitaxial relationship at the RuO₂/TiO₂ interface that should stabilize the RuO₂ species and prevent the agglomeration.

In contrast, the interaction between Ni and TiO₂ is relatively weak, and this is demonstrated by the agglomerated Ni species in 5Ni/TiO₂ during the reduction process. However, with the addition of Ru, the migration of Ni is limited by the Ru obstacles that are stabilized on the TiO₂ support, as shown in the schematic in **Figure 2** (or **Figure R10**). Hence, TiO₂ is a suitable support for the formation of highly dispersed Ru- and Ni-based clusters.

Figure R10. Synthesis and microscopy characterization of catalysts. HAADF-STEM and corresponding EDS-mapping images of O, Ti, Ni, and Ru elements of 2Ru5Ni/TiO₂ after air calcination (a-b) and H₂ reduction (c-d) respectively. (e-f) HAADF-STEM images of reduced 5Ni/TiO₂, the inset image in (f) corresponds to the EDS-mapping image of Ni element. (g) Schematic illustration of the

evolution of bimetallic Ru-Ni and monometallic Ni species on TiO₂ during sequential calcination and hydrogen reduction treatments respectively.

4) The reference paper on RuNi/TiO₂ reported at ChemCatChem 2018, 10, 3526-3531 should be cited and compared.

Response to comments: We thank this reviewer for the suggestion. The reference recommended by the reviewer shows that TiO₂ acts as efficient support for the synthesis of RuNi bimetallic alloy by simple impregnation followed by H₂ reduction at 300 °C. In contrast, this RuNi bimetallic alloy cannot be effectively formed on other types of support (SiO₂, Al₂O₃, ZrO₂, MWCNT, Graphene, and MgO). As a result, the RuNi/TiO₂ catalyst exhibits a remarkably high catalytic activity for the dehydrogenation from ammonia borane compared with those prepared catalysts with other supports.

Notably, in this reference, the content of Ru is higher than that of Ni for RuNi/TiO₂ (Ru:Ni=1:0.3), and the formed alloy is a Ru-rich one (an alloy based on Ru-matrix). In contrast, the scenario in our work is a Ni-rich composition in 2Ru5Ni/TiO₂, and this may introduce phase separation instead of alloying. This result suggests that the Ru/Ni ratios should have an influence on the structure of RuNi species on TiO₂.

Here, this reference has been cited in the revised manuscript, as the reviewer suggested.

Reviewer #4

The manuscript describes the development of bimetallic RuNi@TiO₂ catalysts with the anti-CO-poisoning ability for potential application in a LOHC-based hydrogen storage system, in detail the toluene-MCH pair and the concept was demonstrated in the hydrogenation step of toluene hydrogenation to MCH for H₂ storage. This is an important topic for investigation and the findings from the current work do show the progress beyond the state of the art and provided some insights into the relevant mechanisms of the system under investigation. I am not the expert in all the areas included in the current work, *but after reading it carefully, I identified the following comments/queries for the authors to consider for revision and improvement.*

Response to comments: We thank the reviewer for the very positive view of our work. Per the request of the reviewer, we have added additional experimental results and discussion in the following point-by-point response and the revised manuscript.

1. The title claimed mild conditions, and I think this should be reflected by a revised abstract and conclusion, i.e. < 200 °C at atmospheric pressure, indeed being much lower than the common conditions of 200-300 °C and 10-50 bar.

Response to comments: We appreciate this reviewer's comment. However, the expression "mild" is not that suitable here as toluene hydrogenation using pure H₂ can happen at room temperature (such as *J. Am. Chem. Soc.* **2010**, 132, 6541-6549). Therefore, we revised the title to "CO-tolerant RuNi/TiO₂ Catalyst for Crude Hydrogen Storage and Purification". In addition, we have revised the expression of "mild condition" in the manuscript to practical experimental conditions.

2. Following the comment above, I think the authors need to explain why such high activity was achieved by the current catalytic system at quite short residence time since hydrogenation reactions normally requires elevated P and T.

Response to comments: We thank this reviewer for the suggestion. In fact, for high-pressure batch reactions, toluene hydrogenation took place over Ru-based catalysts even at room temperature (*J. Am. Chem. Soc.* **2010**, 132, 6541-6549). While for flow reactions at atmospheric H₂ pressure, the temperature from 100 – 220 °C is a common range for toluene hydrogenation (*Chemical Engineering Science*, 1993, 48, 3813-3828; *Catal. Today*, **2020**, 356, 64–72).

3. The current catalytic systems using a bubbler for vapor phase reaction, being rather dilute. Can the authors try the conventional packed/trickle bed reactors or batch reactor with gas-liquid systems to show that the developed catalyst is not specific for the microreactor with gas phase reactions?

Response to comments: We thank this reviewer for the comment. Catalytic tests through trickle bed reactor have been added in **Supplementary Figure 7** (or **Figure R11**). Although the toluene feed was unstable by the liquid pump, the result may still tell the feasibility for larger scale application. And the related sentences were added to the manuscript as "For large scale application...in the during the test" marked in blue. In addition, the batch reactor results were also supplied in Supplementary Table 2 (or **Table R3**).

Figure R11. Toluene hydrogenation activity over 2Ru5Ni/TiO₂ on a trickle bed reactor using 0.1 %CO/H₂ at 180 °C. Reaction conditions: 0.1 g catalyst, toluene feed= 0.02 mL/min, 0.1 %CO/H₂ = 50 mL/min, 180 °C, 1 bar. The fluctuation was due to the unstable feeding.

Table R3. Catalytic performance of CO-tolerant hydrogenation reaction on different aromatics using a batch reactor ^a.

Substrates	Conversion	Products and Selectivity			
 b	100 %		100 %		
	100 %		100 %		
	100 %		95.8 %		4.2 %
 b	29.4 %		24.6 %		41.0 %
					34.4 %
	100 %		91.6 %		8.4 %

a. Reaction conditions unless noted otherwise: 30mg 2Ru5Ni/TiO₂, 100 mg substrates, 2 MPa 0.1 %CO/H₂, 3 mL cyclohexane as the solvent, 33 μL n-dodecane as internal standard, 1 h and 180 °C. The total volume of the reactor is 10 mL.

b. 3 mL methyl-cyclohexane as the solvent, instead of cyclohexane.

4. Exemplar GC analysis of products from the rig is needed in the revised SI.

Response to comments: Thanks for the suggestion. Typical GC analysis graph has been added in Supplementary Figure 3 (or **Figure R2**).

5. Deactivation regarding MCH yield was measured during the longevity test, which was not discussed, though comment stating it stayed over 60% was provided. The authors need to elucidate this aspect since maintaining MCH yield is the key target of the work, yet continuous decline of the yield is very obvious.

Response to comments: We thank the comment from the reviewer, and this is a crucial question on the performance of the catalysts. The activity decline during the 24 h continuous reaction did exist. We compare the weight loss of fresh and spent catalysts by TGA (Thermogravimetric Analysis), and the result is shown in **Figure R12**. The weight loss of fresh 2Ru5Ni/TiO₂ was 7 % and the spent catalyst was only about 2 %, which indicates the carbon deposition may not be the reason causing the activity loss during the reaction. However, through our STEM results, we found the growth of Ru and Ni species on the spent 2Ru5Ni/TiO₂ (**Supplementary Figure 10** or **Figure R13**). So the MCH yield drop was probably resulted from the metal particle agglomeration, and the detailed discussion has been added in the manuscript as “However, the slight MCH yield drop... of spent 2Ru5Ni/TiO₂ catalyst” marked in blue.

Figure R12. TGA analysis of fresh and spent 2Ru5Ni/TiO₂.

Figure R13. Representative HAADF-STEM image and the corresponding EDS mapping images of Ni and Ru of spent 2Ru5Ni/TiO₂ after one cycle of 100 – 220 °C CO tolerant toluene hydrogenation reaction; Reaction condition: 0.1 %CO/0.1 %Ar/H₂, GHSV of the carrier gas = 36,000 mL/g_{cat}/h and WHSV of toluene = 2.1 h⁻¹.

6. Regarding the discussion for CO₂ methanation, I think the authors oversell the performance. CO₂ hydrogenation requires much higher temperatures than the current 100-220 °C, therefore, a simple discussion will be fine rather than being surprising about the phenomenon of non CO₂ conversion in the current system.

Response to comments: We appreciate this reviewer’s suggestion. We have revised the statement in the manuscript regarding the catalytic performance of 2Ru5Ni/TiO₂ in CO₂ methanation properly.

7. Generic feature of the develop catalyst is needed, i.e. for hydrogenation of other LOHC pairs such as DBT to PDBT and/or benzene to cyclohexane.

Response to comments: We thank this reviewer for the insightful comment. We have extended the substrates besides toluene to benzene and p-xylene in a gas-solid fixed bed (**Supplementary Figure 8** or **Figure R14**). In addition, the trials of benzene, biphenyl, phenyl and quinolone in batch reactor were displayed in **Supplementary Table 2** (or **Table R3**). Apparently, the 2Ru5Ni/TiO₂ catalyst shows the potential to realize the crude hydrogen storage over these organic substrates. And the sentences were added to the manuscript as “In addition, to prove the developed catalyst...which means the wide

feasibility of the catalyst” marked in blue. For hydrogenation of DBT, the result was listed below (**Table R4**). As the products were hard to be distinguished by GC, so it was not shown in SI.

Figure R14. CO-tolerant aromatics (benzene, toluene and p-xylene) hydrogenation activities over 2Ru5Ni/TiO₂. Reaction conditions: 180 °C, 0.1 %CO/0.1 %Ar/H₂, GHSV of the carrier gas = 36,000 mL/g_{cat}/h, WHSV of benzene = 5.3 h⁻¹, toluene = 2.1 h⁻¹, p-xylene = 1.5 h⁻¹.

Table R4. Catalytic performance of CO-tolerant hydrogenation reaction on DBT using a batch reactor

Substrates	Conversion	Products and Selectivity
	56.1 %	18.9 % others 81.1 %

Reaction conditions: 30 mg 2Ru5Ni/TiO₂, 100 mg DBT, 2 MPa 0.1 %CO/H₂, 3 mL cyclohexane as the solvent, 33 µL n-dodecane as internal standard, 1 h and 180 °C, the total volume of the reactor is 10 mL.

REVIEWERS' COMMENTS

Reviewer #1 (Remarks to the Author):

The authors have satisfactorily answered to my comments, both in the main article and in the supplementary material. Yet, I believe that it would strengthen further the paper if the authors mentioned that no $\text{Ni}(\text{CO})_4$ could be evidence during their experiments. This point is important on two aspects, first to strengthen the experimental data and second, as a safety matter, to stress that the evolution of such highly toxic compound (more toxic than HCN) should always be born in mind when using CO over nickel at relatively low temperatures.

Reviewer #2 (Remarks to the Author):

The authors have made a great revision on this manuscript, the confusions were cleared up and more important experiments were added, which address the loosing point of the mechanism understanding. The reviewer agrees the publication it on nature communications.

Reviewer #3 (Remarks to the Author):

Because moderate modifications were made, this can be accepted.

Reviewer #4 (Remarks to the Author):

The authors have carefully considered all reviewers' comments and revised the manuscript thoroughly with many additional experiments to make the work more solid and interesting. Based on my previous overall positive comments on the work, I recommend publication of the current version of the manuscript.

Response to reviewers

We thank the reviewers for the very positive views and constructive comments. We have addressed all the comments point-by-point and revised the manuscript accordingly. In this response letter, comments from the reviewers are summarized with the major concerns quoted in blue italic.

Reviewer #1

The authors have satisfactorily answered to my comments, both in the main article and in the supplementary material. Yet, I believe that it would strengthen further the paper if the authors mentioned that no Ni(CO)₄ could be evidence during their experiments. This point is important on two aspects, first to strengthen the experimental data and second, as a safety matter, to stress that the evolution of such highly toxic compound (more toxic than HCN) should always be born in mind when using CO over nickel at relatively low temperatures.

Response to comments: We thank the reviewer for the very positive view of our work. We have added the sentence in the manuscript “Notably, since the Ni contents kept unchanged after the test, the potential formation of poisonous Ni(CO)₄ thus could be excluded (Supplementary Table 1).” which is highlighted in blue.

Reviewer #2

The authors have made a great revision on this manuscript, the confusions were cleared up and more important experiments were added, which address the loosing point of the mechanism understanding. The reviewer agrees the publication it on nature communications.

Response to comments: We appreciate the reviewer for the very positive view of our work.

Reviewer #3

Because moderate modifications were made, this can be accepted.

Response to comments: We thank the reviewer for the very positive view of our work.

Reviewer #4

The authors have carefully considered all reviewers' comments and revised the manuscript thoroughly with many additional experiments to make the work more solid and interesting. Based on my previous

overall positive comments on the work, I recommend publication of the current version of the manuscript.

Response to comments: We appreciate the reviewer for the very positive view of our work.